# ZeroSearch: Incentivize the Search Capability of LLMs without Searching

## Abstract

Effective information searching is essential for enhancing the reasoning and generation capabilities of large language models (LLMs). Recent research has explored using reinforcement learning (RL) to improve LLMs' search capabilities by interacting with live search engines in real-world environments. While these approaches show promising results, they face two major challenges: (1) **Uncontrolled Document Quality**: The quality of documents returned by search engines is often unpredictable, introducing noise and instability into the training process. (2) **Prohibitively High API Costs**: RL training requires frequent rollouts, potentially involving hundreds of thousands of search requests, which incur substantial API expenses and severely constrain scalability. To address these challenges, we introduce **ZeroSearch**, a novel RL framework that incentivizes the capabilities of LLMs to use a real search engine with simulated searches during training. Our approach begins with lightweight supervised fine-tuning to transform the LLM into a retrieval module capable of generating both useful and noisy documents in response to a query. During RL training, we employ a curriculum-based rollout strategy that incrementally degrades the quality of generated documents, progressively eliciting the model's reasoning ability by exposing it to increasingly challenging retrieval scenarios. Extensive experiments demonstrate that ZeroSearch effectively incentivizes the search capabilities of LLMs using a 3B LLM as the retrieval module. Remarkably, a 7B retrieval module achieves comparable performance to the real search engine, while a 14B retrieval module even surpasses it. Furthermore, it generalizes well across both base and instruction-tuned models of various parameter sizes and is compatible with a wide range of RL algorithms.

## 1 Introduction

Large Language Models (LLMs) Taylor et al. (2022); Chowdhery et al. (2022); Zhao et al. (2023) have demonstrated remarkable performance across a wide range of downstream tasks, including mathematical reasoning, question answering, and code generation Xia et al. (2024); Yamauchi et al. (2023); Imani et al. (2023); Lewkowycz et al. (2022). However, the knowledge encoded in these models is inherently static, constrained by the scope of the data encountered during pretraining. As a result, LLMs remain prone to generating hallucinated content or outdated information Ji et al. (2023); Shuster et al. (2021); Zhang et al. (2023), which undermines their reliability in practical applications. Therefore, it is essential to enable LLMs to access external sources of information to produce more accurate and grounded responses.

One widely adopted approach to addressing this issue is Retrieval-Augmented Generation (RAG), which incorporates external knowledge into the generation pipeline Ram et al. (2023); Shi et al. (2023); Rashkin et al. (2021); Gao et al. (2022); Bohnet et al. (2022); Menick et al. (2022). Early work in this area focused on prompt-based strategies that guide LLMs through query generation, query decomposition, and multi-turn information retrieval Yu et al. (2022); Press et al. (2022); Yoran et al. (2023); Jiang et al. (2023); Shi et al. (2023); Li et al. (2025a). While effective, these strategies often require meticulous prompt engineering and place high demands on the model's reasoning capabilities. To improve the efficiency, subsequent research explored supervised fine-tuning (SFT) to enhance the performance of smaller LLMs Asai et al. (2023); Li et al. (2024a); Jeong et al. (2024). Further advances have focused on test-time scaling techniques Li et al. (2024b); Jiang et al. (2024b); Zhao et al. (2024); Jiang et al. (2024a), such as Monte Carlo Tree Search (MCTS), which dynamically

expands the search space during inference. Although promising, such methods incur significant computational overhead, posing challenges for practical deployment.

Recently, reinforcement learning (RL) has emerged as a promising strategy for further improving LLM performance by enhancing their reasoning and decision-making capabilities Guo et al. (2025); Hou & et al. (2025). Notably, RL-based models such as OpenAI-o1 and DeepSeek-R1 have demonstrated substantial gains in logical inference and iterative reasoning—achieved purely through reward-driven learning, without relying on explicit step-by-step supervision Kumar et al. (2024). Within this paradigm, several studies have explored using RL to train policy models that can more effectively search for relevant information. Representative examples include Search-R1 Jin et al. (2025), R1-Searcher Song et al. (2025), and ReSearch Kumar & et al. (2025). Notably, DeepResearcher Zheng et al. (2025) and WebThinker Li et al. (2025b) introduce live interaction with commercial search engines such as Google, allowing models to train in an environment that closely mirrors real-world web search. Despite these advancements, integrating RL with real-world search scenarios presents several significant challenges: (1) **Uncontrolled Document Quality**: The quality of documents retrieved from live search engines is often unpredictable, introducing noise and instability into the training process. (2) **Prohibitively High API Costs**: RL training requires frequent rollouts, potentially involving hundreds of thousands of API calls, which incur substantial financial costs and severely limit scalability.

To address these challenges, we propose **ZEROSEARCH**—a novel reinforcement learning framework that enables LLMs to learn search strategies without interacting with real search engines. Our key insight is that **LLMs have acquired extensive world knowledge during large-scale pretraining and are capable of generating relevant documents given a search query** Yu et al. (2022). The primary difference between a real search engine and a simulation LLM lies in the textual style of the returned content. However, with lightweight supervised fine-tuning, even relatively small LLMs can effectively simulate the behavior of real search engines. In addition to eliminating API costs, an important advantage of using LLMs for document generation is the ability to control document quality. Specifically, during supervised fine-tuning, documents that lead to correct or incorrect answers are distinguished through prompt design, enabling the simulation LLM to learn to generate either useful or noisy documents simply by adjusting a few words in the prompt. Building on this, **we introduce a curriculum rollout mechanism during training**, in which the quality of the generated documents is gradually degraded over time to simulate increasingly challenging retrieval scenarios. This allows the policy model to first learn basic output formats and task requirements before progressively adapting to more challenging and noisy retrieval scenarios. More importantly, ZEROSEARCH exhibits strong scalability: increasing the number of GPUs significantly accelerates the generation throughput of the simulation LLM, thereby enabling efficient large-scale rollout. Empirical results show that **even a 3B LLM used as the simulated search engine can effectively incentivize the policy model's capabilities to utilize a real search engine**. When a 7B LLM is used as the simulator, the resulting policy model achieves performance comparable to that trained with a real search engine, while a 14B simulator even leads to better performance. ZEROSEARCH is compatible with both base and instruction-tuned models of various parameter sizes, removing the need for separate supervised warm-up stages. Moreover, it integrates seamlessly with widely used RL algorithms, including REINFORCE Williams (1992), Proximal Policy Optimization (PPO) Schulman et al. (2017), and Group Relative Policy Optimization (GRPO) Shao et al. (2024); Guo et al. (2025).

Our contributions can be summarized as follows:

- We propose ZEROSEARCH, a novel reinforcement learning framework that incentivizes the capability of LLMs to use real search engines without interacting with them during training.

- Through supervised fine-tuning, we transform the LLM into a retrieval module capable of generating both useful and noisy documents in response to a query. We further introduce a curriculum rollout mechanism to progressively elicit the model's reasoning ability by exposing it to increasingly challenging retrieval scenarios.

- We conduct extensive experiments on both in-domain and out-of-domain datasets. Results show that ZEROSEARCH outperforms real search engine-based models while incurring zero API cost. Moreover, it generalizes well across both base and instruction-tuned LLMs of various parameter sizes and supports different reinforcement learning algorithms.

## 2 RELATED WORK

### 2.1 RETRIEVAL-AUGMENTED GENERATION

Retrieval-augmented generation (RAG) enhances generation performance by integrating relevant external knowledge into the generation pipeline. Early research primarily adopted prompt-based approaches, guiding LLMs through processes such as query generation, query decomposition, and multi-turn information retrieval Yu et al. (2022); Press et al. (2022); Yoran et al. (2023); Jiang et al. (2023); Shi et al. (2023); Li et al. (2025a). Despite their effectiveness, these methods often require intricate prompt engineering and impose substantial demands on the model's reasoning capabilities. To improve efficiency and reduce dependency on strong black-box LLMs, subsequent work has proposed supervised fine-tuning strategies for smaller LLMs. For instance, Self-RAG Asai et al. (2023) employs a self-reflection mechanism, iteratively refining model outputs through predicted reflection tokens. RetroLLM Li et al. (2024a) integrates retrieval and generation by enabling the model to directly generate fine-grained evidence from the corpus via constrained decoding. Recent advances also include test-time scaling techniques Li et al. (2024b); Jiang et al. (2024b); Zhao et al. (2024); Jiang et al. (2024a), notably Monte Carlo Tree Search (MCTS), which dynamically expands the search space during inference. For example, RAG-star Jiang et al. (2024a) integrates retrieved information into a tree-based reasoning process, while AirRAG Feng et al. (2025) employs MCTS to activate intrinsic reasoning capabilities and expand the solution space. Despite promising results, these approaches introduce significant computational overhead, limiting their practical applicability.

### 2.2 LEARNING TO SEARCH THROUGH REINFORCEMENT LEARNING

Recently, reinforcement learning (RL) has emerged as a promising paradigm for enhancing the reasoning capabilities of LLMs Guo et al. (2025); Hou & et al. (2025). Notable RL-based models such as OpenAI-o1 and DeepSeek-R1 have demonstrated remarkable capabilities in logical inference and iterative reasoning, purely driven by reward signals without explicit step-by-step supervision Kumar et al. (2024). Several studies have also explored RL techniques specifically designed to train models for effective information retrieval. For instance, Search-R1 Jin et al. (2025) employs reinforcement learning to autonomously generate multiple search queries during step-by-step reasoning. Similarly, R1-Searcher Song et al. (2025) proposes a two-stage, outcome-based RL method aimed at enhancing search capabilities. ReSearch Kumar & et al. (2025) leverages RL to teach models to reason through searches, entirely without supervision on intermediate reasoning steps. However, these methods usually employ static, local textual corpora such as Wikipedia and fail to capture the complexities of real-world interaction. To bridge this gap, DeepResearcher Zheng et al. (2025) and WebThinker Li et al. (2025b) introduce direct interaction with commercial search engines such as Google, allowing training environments that closely approximate real-world search scenarios. While achieving superior performance, these real-time retrieval methods face significant challenges, including unpredictable document quality, prohibitive high API costs that adversely affect system scalability. To address these limitations, we propose ZEROSEARCH, a method leveraging an LLM to simulate real-time search, effectively removing dependence on costly, rate-limited real search APIs. Through lightweight supervised fine-tuning, ZEROSEARCH allows explicit control over document quality and implements a curriculum rollout mechanism that enhances training stability and robustness.

## 3 ZEROSEARCH

In this section, we first formalize the reinforcement learning objective without a search engine. We then detail the design of ZEROSEARCH, covering the training template, search simulation tuning, curriculum-based rollout strategy, reward design, and training algorithms.

### 3.1 REINFORCEMENT LEARNING WITHOUT A SEARCH ENGINE

We propose a reinforcement learning framework that eliminates the need for a real search engine by leveraging an LLM to simulate the search engine. The optimization objective is formulated as:

$$\max_{\pi_\theta} \mathbb{E}_{x \sim \mathcal{D}, \, y \sim \pi_\theta(\cdot | x; \pi_\psi)} \big[ r_\phi(x, y) \big] \; - \; \beta \, \mathrm{D_{KL}} \big[ \pi_\theta(y \mid x; \pi_\psi) \, \big\| \, \pi_{\mathrm{ref}}(y \mid x; \pi_\psi) \big],$$

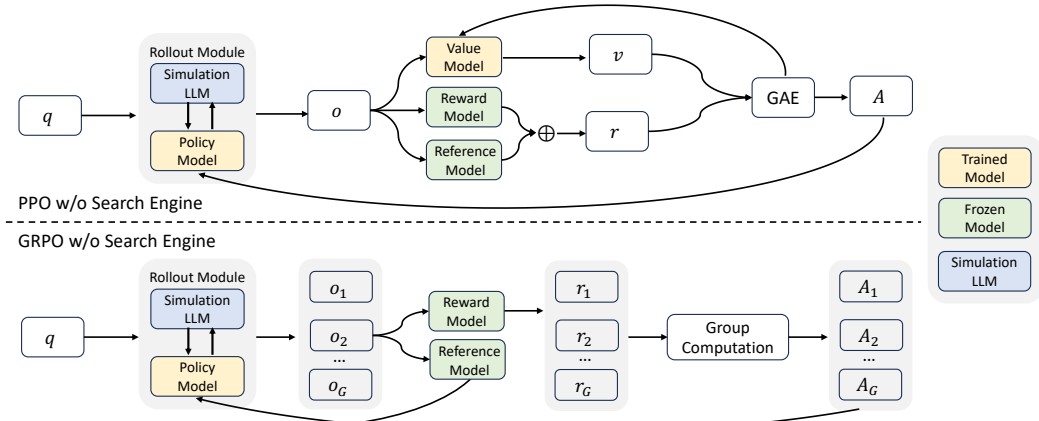

Figure 1: Demonstration of PPO and GRPO training without the search engine.

$$\max_{\pi_\theta} \quad \mathbb{E}_{x \sim \mathcal{D}, y \sim \pi_\theta(\cdot|x;\pi_\psi)}[r_\phi(x, y)]$$
$$-\beta D_{\mathrm{KL}}(\pi_\theta(y|x)||\pi_{\mathrm{ref}}(y|x; \pi_\psi))$$

where $\pi_\theta$ is the policy model to be optimized, $\pi_{\mathrm{ref}}$ is the reference model, and $r_\phi$ denotes the reward function. $\pi_\psi$ represents the simulation LLM, whose parameters remain fixed throughout training.

## 3.2 TRAINING TEMPLATE

---

Answer the given question. You must conduct reasoning inside `<think>` and `</think>` first every time you get new information. After reasoning, if you find you lack some knowledge, you can call a search engine by `<search>` query `</search>`, and it will return the top searched results between `<information>` and `</information>`. You can search as many times as you want. If you find no further external knowledge needed, you can directly provide the answer inside `<answer>` and `</answer>` without detailed illustrations. For example, `<answer>` Beijing `</answer>`. Question:

---

Table 1: Training template. The question is appended at the end during training and inference.

In ZEROSEARCH, rather than relying on supervised fine-tuning for generation, we follow Jin et al. (2025) and apply a multi-turn interaction template that guides the policy model through iterative reasoning and information retrieval until a final answer is reached.

As illustrated in Table 1, the interaction is divided into three distinct stages: First, the model articulates its internal reasoning within the `<think>...</think>` tag. Second, if additional evidence is needed, it issues a search query within the `<search>...</search>` tag. Finally, once sufficient information has been retrieved, the model provides its answer in the `<answer>...</answer>` tag. This explicit separation of reasoning, searching, and answering enforces a structured decision-making process, enhancing the model's transparency and reliability.

## 3.3 SEARCH SIMULATION TUNING

During rollout, we use the LLM to simulate a real search engine by generating documents in response to queries. A straightforward approach is to directly prompt the LLM to generate documents. However, this often results in a noticeable style gap compared to outputs from real search engines.

To bridge this gap, we propose a lightweight supervised fine-tuning (SFT) procedure. Specifically, we first collect interaction trajectories by prompting the LLM to engage with a real search engine in a multi-turn manner until a final answer is reached. From these trajectories, we extract query-document

pairs and employ the LLM as the judge to independently assess whether each document contains sufficient information to answer the corresponding query. If the answer is affirmative, the document is labeled as a useful output; otherwise, it is labeled as a noisy output.

Then, we perform lightweight SFT to enhance the LLM's ability to generate both useful and noisy outputs in response to queries. As shown in Table 2, the distinction between useful and noisy outputs can be effectively controlled by adjusting a few words in the prompt. Besides, we also incorporate the input question and its corresponding answer into the prompt to broaden the knowledge boundary of the simulation LLM. After fine-tuning, the simulation LLM is capable of generating both useful and noisy documents, enabling dynamic document quality control during rollout.

### 3.4 ROLLOUT WITH CURRICULUM SEARCH SIMULATION

---

You are the Google search engine.
Given a query, you need to generate five [useful / noisy] documents for the query.
The user is trying to answer the question: [question] whose answer is [ground truth].
Each document should contain about 30 words, and these documents should contain [useful / noisy] information.
Query: [query]
[Useful / Noisy] Output:

---

Table 2: Template for Search Simulation. The useful and noisy keywords are used to control the quality of the generated documents. The input question and its ground-truth answer are also included in the prompt to help extend the simulation LLM's knowledge coverage.

During rollout, the policy model performs interactive reasoning and generates search queries, which are fed into the simulation LLM to produce corresponding documents. To gradually increase the difficulty of training, we introduce a curriculum learning-based rollout mechanism, where the quality of the generated documents is progressively degraded over time. This is controlled by a probability function $p_i$ that governs the likelihood of generating noisy documents at step $i$:

$$p_i = p_s + \frac{b^{i/m} - 1}{b - 1}(p_e - p_s) \tag{1}$$

Here, $p_s$ and $p_e$ represent the initial and final noise probabilities, $i$ and $m$ denote the current and total number of training steps, and $b$ is the exponential base, with a default value of 4. As training progresses, the ratio $i/m$ increases, leading to a higher $p_i$ value—i.e., a greater chance of producing noisy documents. This allows the policy model to first learn basic output structures and task requirements, before gradually adapting to more challenging and noisy retrieval scenarios.

### 3.5 REWARD DESIGN

The reward signal serves as the primary supervision in the reinforcement learning process. In this work, we adopt a rule-based reward function that focuses solely on answer accuracy. During preliminary experiments, we observed that using exact match (EM) as the reward metric often led to reward hacking: the policy model tended to produce excessively long answers to increase the chance of including the correct answer. To mitigate this issue, we adopt an F1 score-based reward, which balances precision and recall, and is calculated as:

$$r_\phi(x, y) = \frac{2 \times IN}{PN + RN},$$

where *IN* denotes the number of overlapping words between the prediction and the ground truth, *PN* is the number of words in the prediction, and *RN* is the number of words in the ground truth. We do not incorporate an additional reward for output format, as we observe that the model consistently produces well-formed responses without explicit supervision.

## 3.6 TRAINING ALGORITHM

Our approach is compatible with a wide range of reinforcement learning algorithms, including REINFORCE Williams (1992), Proximal Policy Optimization (PPO) Schulman et al. (2017), and Group Relative Policy Optimization (GRPO) Shao et al. (2024); Guo et al. (2025), each offering distinct advantages for optimizing retrieval-augmented reasoning.

In ZEROSEARCH, the rollout sequence comprises both tokens generated by the policy model and document tokens returned by the simulation LLM. Applying the same optimization procedure uniformly across both types of tokens can lead to training instability, as the document tokens are externally generated and not directly controlled by the policy model.

To mitigate this, we introduce a loss masking mechanism for document tokens, ensuring that gradients are only computed with respect to the model's own outputs. This strategy stabilizes the RL training process while preserving the effectiveness of retrieval-augmented generation.

## 4 MAIN RESULTS

### 4.1 DATASETS AND EVALUATION METRICS

We evaluate ZEROSEARCH on a diverse set of question answering benchmarks: (1) **Single-Hop Question Answering**, including NQ Kwiatkowski et al. (2019), TriviaQA Joshi et al. (2017), and PopQA Mallen et al. (2022). (2) **Multi-Hop Question Answering**, including HotpotQA Yang et al. (2018), 2WikiMultiHopQA Ho et al. (2020), Musique Trivedi et al. (2022), and Bamboogle Press et al. (2022).

We follow Jin et al. (2025) and adopt Exact Match (EM) as our evaluation metric. A prediction is deemed correct if its normalized form exactly matches any of the normalized ground-truth answers.

### 4.2 BASELINES

To evaluate the effectiveness of ZEROSEARCH, we compare our method with the following baselines. (1) **Vanilla Prompting Methods**: This category includes direct prompting, Chain-of-Thought (CoT), and standard Retrieval-Augmented Generation (RAG). (2) **Advanced RAG Methods**: We consider RAgent Li et al. (2025a) and Search-o1 Li et al. (2025a), which iteratively search for relevant information. (3) **RL Tuning Methods**: This category includes R1 and Search-R1 Jin et al. (2025). In R1, the policy model is trained to perform in-depth reasoning based solely on its internal knowledge. In contrast, Search-R1 enables the policy model to interact with a real search engine multiple times during inference.

To ensure a fair comparison, we adopt the F1 score as the reward metric across all RL methods. Notably, among RL-based search baselines, **we compare only with Search-R1, as it avoids complex reward design, data selection, or elaborate training pipelines.** This setting allows for a direct and equitable comparison between the real search engine and our simulated search engine.

### 4.3 EXPERIMENTAL SETUP

We conduct experiments using three model families: Qwen-2.5-7B (Base/Instruct) and Qwen-2.5-3B (Base/Instruct) Yang et al. (2024), as well as LLaMA-3.2-3B (Base/Instruct) Dubey et al. (2024). To simulate real-world retrieval scenarios, we utilize Google Web Search via the SerpAPI[1] as the external search engine. **During evaluation, all methods use SerpAPI as the search engine to ensure a fair comparison.** The number of retrieved documents is fixed at five across all methods to ensure a fair comparison.

For datasets, following the setup in Jin et al. (2025), we merge the training sets of NQ and HotpotQA to create a unified dataset for all fine-tuning-based approaches. Evaluation is conducted on seven datasets to assess both in-domain and out-of-domain performance. For prompt-based baselines, we use Instruct models, as Base models typically struggle to follow task instructions. For RL-based methods, we evaluate both Base and Instruct variants to assess generality across model types.

---

[1] https://serpapi.com/

| Method | Single-Hop QA | | | Multi-Hop QA | | | | |
|---|---|---|---|---|---|---|---|---|
| | NQ | TriviaQA | PopQA | HotpotQA | 2Wiki | Musique | Bamboogle | Avg. |
| *Qwen-2.5-7B-Base/Instruct* | | | | | | | | |
| Direct Answer | 11.60 | 35.60 | 1.20 | 16.40 | 22.20 | 4.80 | 14.40 | 15.17 |
| CoT | 12.80 | 35.60 | 3.80 | 16.20 | 22.60 | 6.60 | 24.00 | 17.37 |
| RAG | 27.40 | 58.20 | 17.80 | 25.80 | 23.20 | 9.40 | 16.80 | 25.51 |
| RA-Agent | 21.20 | 40.20 | 8.80 | 19.60 | 19.60 | 7.60 | 28.00 | 20.71 |
| Search-o1 | 19.40 | 40.60 | 11.40 | 17.00 | 27.00 | 8.60 | 30.40 | 22.06 |
| R1-base | 27.60 | 47.40 | 27.40 | 21.00 | 29.20 | 9.80 | 27.78 | 27.17 |
| R1-instruct | 27.00 | 45.80 | 24.20 | 21.60 | 27.80 | 8.40 | 25.00 | 25.69 |
| Search-R1-base | 43.40 | 61.40 | 54.60 | 31.20 | **37.20** | 18.20 | 30.56 | 39.51 |
| Search-R1-inst | 42.40 | 63.40 | 51.60 | 32.80 | 33.20 | 17.40 | 26.39 | 38.17 |
| ZEROSEARCH-base | 42.40 | **66.40** | 60.40 | 32.00 | 34.00 | 18.00 | **33.33** | 40.93 |
| ZEROSEARCH-inst | **43.60** | 65.20 | 48.80 | **34.60** | 35.20 | **18.40** | 27.78 | 39.08 |
| *Qwen-2.5-3B-Base/Instruct* | | | | | | | | |
| Direct Answer | 12.40 | 30.60 | 5.60 | 16.00 | 19.20 | 4.40 | 16.80 | 15.00 |
| CoT | 15.00 | 33.60 | 3.60 | 16.20 | 18.00 | 3.60 | 12.80 | 14.69 |
| RAG | 31.60 | 58.00 | 15.20 | 24.20 | 23.20 | 8.20 | 15.20 | 25.09 |
| RA-Agent | 15.20 | 28.40 | 6.60 | 12.60 | 16.60 | 2.60 | 13.60 | 13.66 |
| Search-o1 | 16.60 | 31.00 | 8.20 | 14.80 | 22.40 | 5.20 | **22.40** | 17.23 |
| R1-base | 14.20 | 34.80 | 20.80 | 19.60 | 28.40 | 6.40 | 5.56 | 18.54 |
| R1-instruct | 19.80 | 33.00 | 19.40 | 19.40 | 26.40 | 4.40 | 11.11 | 19.07 |
| Search-R1-base | 40.60 | 60.00 | 44.20 | 29.20 | 32.00 | 11.20 | 12.50 | 32.81 |
| Search-R1-inst | 35.80 | 55.80 | 26.00 | 33.20 | 26.00 | 7.60 | 12.50 | 28.13 |
| ZEROSEARCH-base | **43.00** | **61.60** | 41.40 | **33.80** | **34.60** | **13.00** | 13.89 | **34.47** |
| ZEROSEARCH-inst | 41.40 | 57.40 | **44.80** | 27.40 | 30.00 | 9.80 | 11.11 | 31.70 |
| *LLaMA-3.2-3B-Base/Instruct* | | | | | | | | |
| Direct Answer | 16.20 | 29.60 | 7.40 | 12.60 | 9.20 | 2.00 | 8.00 | 12.14 |
| CoT | 26.20 | 44.40 | 2.80 | 16.00 | 10.20 | 5.80 | 21.60 | 18.14 |
| RAG | 30.00 | 57.60 | 26.40 | 23.40 | 17.60 | 9.60 | 11.20 | 25.11 |
| RA-Agent | 22.40 | 36.20 | 11.40 | 16.60 | 21.00 | 5.60 | 26.40 | 19.94 |
| Search-o1 | 24.20 | 48.40 | 8.80 | 19.40 | 17.40 | 6.00 | **32.00** | 22.31 |
| R1-base | 28.40 | 44.20 | 30.00 | 22.80 | 28.40 | 7.00 | 11.11 | 24.56 |
| R1-instruct | 35.00 | 52.20 | 27.60 | 21.60 | 17.80 | 11.40 | 20.83 | 26.63 |
| Search-R1-base | 41.20 | 60.00 | 44.00 | 29.60 | 31.60 | 13.60 | 19.44 | 34.21 |
| Search-R1-inst | 37.60 | 53.60 | 44.20 | 21.00 | 20.40 | 8.80 | 27.78 | 30.48 |
| ZEROSEARCH-base | **43.40** | **63.80** | **48.40** | **32.20** | **35.60** | **13.80** | 15.28 | **36.07** |
| ZEROSEARCH-inst | 40.20 | 58.00 | 46.00 | 22.80 | 21.40 | 10.40 | 18.06 | 30.98 |

Table 3: Main results using different LLMs as the backbone. The best performance is set in bold.

During experimentation, we deploy the simulation server on 4 H20 GPUs and conduct RL training on another 4 H20 GPUs. To train ZEROSEARCH, we experiment with three RL algorithms: REIN-FORCE, GRPO, and PPO. Unless otherwise specified, all RL-based methods use REINFORCE as the default training algorithm. The fine-tuned Qwen-2.5-14B-Instruct model serves as the default simulated search engine across all experiments. **During inference, all models interact with the real web environment via Google Web Search to ensure a fair comparison.** For further implementation details, including complete hyperparameter configurations, please refer to Section G.

## 4.4 PERFORMANCE

Table 3 presents a comparison between ZEROSEARCH and several baseline methods across seven datasets. Based on the results, several key observations can be drawn:

**ZEROSEARCH consistently outperforms all baseline methods.** This performance advantage holds for both in-domain datasets (*i.e.*, NQ and HotpotQA) and out-of-domain datasets (*i.e.*, TriviaQA, PopQA, 2WikiMultiHopQA, Musique, and Bamboogle), demonstrating the robustness of our method.

**ZEROSEARCH surpasses methods that rely on real search engines.** Compared to Search-R1, which utilizes the real search engine, ZEROSEARCH achieves better performance, highlighting its potential as an effective alternative to real search engines in large-scale reinforcement learning.

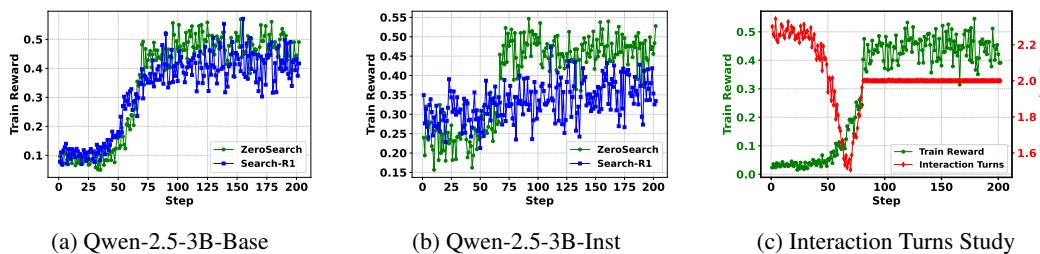

| (a) Qwen-2.5-3B-Base | (b) Qwen-2.5-3B-Inst | (c) Interaction Turns Study |

Figure 2: (a-b): Reward curve comparison between ZEROSEARCH and Search-R1 using Qwen-2.5-3B. (c): The reward curve and interaction turns during training of LLaMA-3.2-3B-Base.

| Search Engine | NQ | TriviaQA | PopQA | HotpotQA | 2Wiki | Musique | Bamboogle | Avg. |
|---|---|---|---|---|---|---|---|---|
| Base Model | 12.40 | 19.40 | 11.20 | 4.40 | 6.80 | 2.00 | 5.56 | 8.82 |
| Prompt-3B | 35.80 | 56.00 | 42.20 | 25.60 | 27.00 | 4.20 | 15.28 | 29.44 |
| Prompt-7B | 38.40 | 59.40 | 43.40 | 27.80 | 30.00 | 11.00 | 9.72 | 31.39 |
| Prompt-14B | 40.00 | 55.40 | 42.00 | 26.80 | 33.20 | 10.60 | 12.50 | 31.50 |
| SFT-3B | 37.00 | 55.20 | 43.00 | 30.20 | 28.20 | 10.00 | 9.72 | 30.47 |
| SFT-7B | 40.40 | 60.40 | 45.40 | 31.20 | 30.80 | 12.60 | 13.89 | 33.53 |
| SFT-14B | 43.00 | 61.60 | 41.40 | 33.80 | 34.60 | 13.00 | 13.89 | 34.47 |
| Google | 40.60 | 60.00 | 44.20 | 29.20 | 32.00 | 11.20 | 12.50 | 32.81 |

Table 4: Performance of different simulated search engines with Qwen-2.5-3B-Base as the policy model. The Base Model refers to performance without any RL training.

**ZEROSEARCH demonstrates strong generalizability.** Across different model families, parameter sizes, and types (*i.e.*, base or instruction-tuned), ZEROSEARCH consistently outperforms baselines. Moreover, its performance further improves with larger models, highlighting its scalability.

## 5 FURTHER ANALYSIS

### 5.1 COMPARE ZEROSEARCH WITH REAL SEARCH ENGINE

We compare the reward curves of ZEROSEARCH and Search-R1 (using a real search engine) on Qwen-2.5-3B, as shown in Figures 2a and 2b. Several key observations can be made:

**The overall reward trends are similar across both methods.** As training progresses, the reward scores of both ZEROSEARCH and Search-R1 steadily increase, indicating that the policy models in both settings effectively learn to interact with search engines and produce correct answers.

**ZEROSEARCH achieves a more pronounced reward improvement.** As shown in Figure 2b, ZEROSEARCH initially lags behind Search-R1 but eventually surpasses it with less fluctuation, thanks to the curriculum rollout mechanism that helps the model gradually master search tool usage.

**ZEROSEARCH generalizes well across both base and instruction-tuned models.** Under both model types, ZEROSEARCH steadily improves reward performance, underscoring its generalizability.

### 5.2 CHOICE OF SIMULATION LLMS

In this section, we investigate how different simulation engine configurations affect the performance of the policy model, including prompt-based and fine-tuned LLMs ranging from 3B to 14B parameters. Based on the results in Table 4, we make the following observations:

First, the fine-tuned 7B simulation engine (SFT-7B) achieves performance comparable to that of Google Search, while the 14B variant (SFT-14B) even surpasses it. This demonstrates the feasibility of using a well-trained LLM as a substitute for real search engines in reinforcement learning setups.

| Method | NQ | TriviaQA | PopQA | HotpotQA | 2Wiki | Musique | Bamboogle | Avg. |
|--------|-----|----------|-------|----------|-------|---------|-----------|------|
| ***Qwen-2.5-3B-Base*** | | | | | | | | |
| Curriculum | 43.00 | 61.60 | 41.40 | 33.80 | 34.60 | 13.00 | 13.89 | 34.47 |
| Random | 41.40 | 59.00 | 44.20 | 29.00 | 31.40 | 10.60 | 12.50 | 32.59 |
| ***LLaMA-3.2-3B-Base*** | | | | | | | | |
| Curriculum | 43.40 | 63.80 | 48.40 | 32.20 | 35.60 | 13.80 | 15.28 | 36.07 |
| Random | 40.40 | 62.80 | 49.60 | 29.80 | 36.00 | 14.20 | 11.11 | 34.84 |

Table 5: Curriculum Rollout Study. We compare the performance of standard and random rollout settings using the Qwen-2.5-3B-Base and LLaMA-3.2-3B-Base as the policy models.

Second, the fine-tuned simulation engines significantly outperform prompt-based simulation engines. Although prompt-based methods are explicitly guided to mimic the response style of a real search engine, a substantial distribution gap remains, leading to inferior performance.

Third, performance improves consistently with increasing model size. Larger simulation LLMs not only exhibit stronger simulation capabilities but also more accurately distinguish between useful and noisy documents, thereby enabling more effective curriculum rollout during training.

### 5.3 INTERACTION TURNS STUDY

In this section, we analyze the training dynamics of ZEROSEARCH by examining both the reward curve and the number of interaction turns throughout the training, using the LLaMA-3.2-3B-Base as the policy model. The results are shown in Figure 2c.

During the early phase of training, the number of interaction turns drops sharply, while the reward increases slowly. This is primarily because the policy model initially lacks knowledge of how to properly invoke the search engine, resulting in redundant interactions. However, it quickly learns the correct format and begins to eliminate unnecessary steps effectively.

As training progresses, both the number of interaction turns and the reward curve increase sharply and then stabilize. This is primarily because the policy model becomes capable of effectively retrieving relevant documents and ultimately achieving correct answers, resulting in higher rewards. Notably, although the reward appears stable in the later stages of training, the underlying task difficulty continues to rise due to the curriculum rollout mechanism. Therefore, the policy must continuously refine its strategy and improve its reasoning ability to maintain the high reward score.

### 5.4 CURRICULUM ROLLOUT STUDY

In this section, we investigate the effectiveness of our curriculum rollout strategy by comparing it against a baseline random setup, where the probability of generating noisy documents is fixed at 0.5 throughout training. The comparison results are summarized in Table 5.

The results clearly indicate that the standard easy-to-hard curriculum consistently outperforms the random rollout variant across both models, verifying the effectiveness of curriculum rollout in our framework. Starting with better search results allows the policy model to first learn how to invoke the search engine and understand the basic output format. As training progresses, the model is exposed to increasingly challenging scenarios, fostering stronger reasoning capabilities.

## 6 CONCLUSION

In this paper, we propose ZEROSEARCH, a novel RL framework that enhances the search capabilities of LLMs without interacting with real search engines. Through supervised fine-tuning, the LLM is transformed into a retrieval module capable of generating both useful and noisy documents. A curriculum rollout mechanism is employed to progressively improve reasoning by exposing the model to increasingly challenging retrieval scenarios. Experimental results show that ZEROSEARCH outperforms real search-based models, generalizes well across both base and instruction-tuned LLMs of varying sizes, and supports a wide range of RL algorithms.

## ETHICS STATEMENT

This work introduces a novel reinforcement learning (RL) training framework that enhances the search capabilities of large language models (LLMs) without requiring interaction with real-world search engines. We believe this approach can significantly reduce the cost of RL training and encourage future research that leverages LLMs as simulators of the real world, potentially yielding positive social impacts. We do not foresee any negative societal impacts arising from this work.

## REPRODUCIBILITY STATEMENT

To ensure the reproducibility of our research, we have included all necessary materials in the supplementary files. This includes the source code for our models and experiments, details of the experimental setup and required datasets. We believe these resources will enable researchers to fully reproduce our findings.

## LLM USAGE DISCLOSURE

The authors utilized LLMs to assist in drafting and refining this manuscript. The authors have thoroughly reviewed and edited the text, and assume full responsibility for the final content, including its accuracy and all claims made.

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

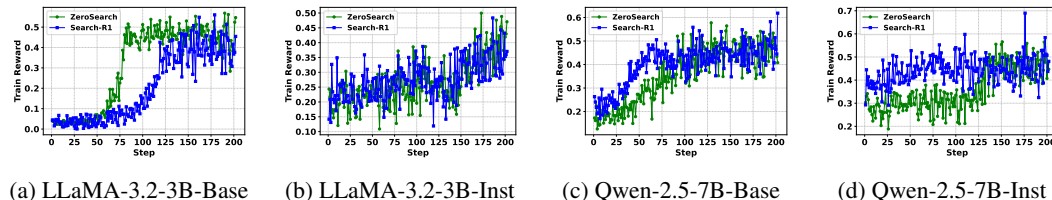

(a) LLaMA-3.2-3B-Base    (b) LLaMA-3.2-3B-Inst    (c) Qwen-2.5-7B-Base    (d) Qwen-2.5-7B-Inst

Figure 3: Reward curve comparison between ZEROSEARCH and Search-R1(using a real search engine).

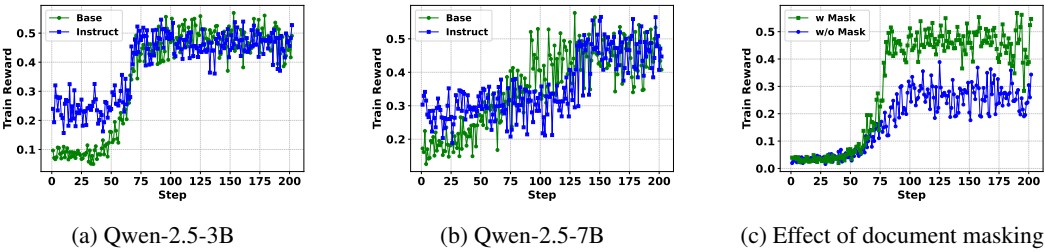

(a) Qwen-2.5-3B      (b) Qwen-2.5-7B      (c) Effect of document masking

Figure 4: (a-b) We compare the reward curve between base and instruct models using Qwen-2.5-3B and Qwen-2.5-7B models. (c): We study the effects of document token loss masking using LLaMA-3.2-3B-base.

## A    COMPARE ZEROSEARCH WITH REAL SEARCH ENGINE

In this section, we present additional results comparing ZEROSEARCH with a real search engine using the LLaMA-3.2-3B and Qwen-2.5-7B model series in Figure 3.

Across both model sizes, ZEROSEARCH consistently achieves a smoother reward curve compared to the real search engine. This is primarily because the quality of documents returned by the real search engine is uncontrollable during rollout. In the early stages of training, low-quality documents may prevent the policy model from developing a correct understanding of the task. In later stages, if the documents are too high-quality, the policy model may not be sufficiently challenged to continue improving its reasoning capability. In contrast, ZEROSEARCH enables dynamic control over document difficulty throughout training. This allows the policy model to first build a foundational understanding of the task and then gradually adapt to more complex scenarios.

## B    COMPARE BASE AND INSTRUCT LLMS

In this section, we compare the training reward curves of base and instruction-tuned models using Qwen-2.5-3B and Qwen-2.5-7B. The results are presented in Figures 4a and 4b.

As shown, instruction-tuned models initially achieve higher rewards, owing to their stronger instruction-following capabilities, which allow them to invoke the search engine more effectively in the early stages of training. As training progresses, both base and instruction-tuned models demonstrate steady reward improvements. Notably, base models demonstrate greater reward improvements and reach performance levels comparable to their instruction-tuned counterparts. These results underscore the compatibility of ZEROSEARCH with both base and instruction-tuned models. Furthermore, they demonstrate that base models can effectively acquire search capabilities through reinforcement learning without the need for supervised fine-tuning as a warm-up.

## C    DIFFERENT RL ALGORITHMS: REINFORCE VS. GRPO VS. PPO

In this section, we evaluate the performance of three widely used RL training algorithms, REINFORCE, GRPO and PPO, within the ZEROSEARCH framework, using the Qwen-2.5-3B-Base as the policy model. The results of this comparison are presented in Table 6.

| Method | NQ | TriviaQA | PopQA | HotpotQA | 2Wiki | Musique | Bamboogle | Avg. |
|--------|-----|----------|-------|----------|-------|---------|-----------|------|
| REINFORCE | 43.00 | 61.60 | 41.40 | 33.80 | 34.60 | 13.00 | 13.89 | 34.47 |
| GRPO | 39.40 | 55.80 | 39.00 | 29.80 | 32.20 | 12.40 | 23.61 | 33.17 |
| PPO | 38.20 | 58.60 | 40.40 | 27.20 | 33.80 | 13.80 | 16.67 | 32.67 |

Table 6: Performance of ZEROSEARCH under different RL algorithms. We compare REINFORCE, GRPO and PPO using Qwen-2.5-3B-Base as the policy model.

| Method | NQ | TriviaQA | PopQA | HotpotQA | 2Wiki | Musique | Bamboogle | Avg. |
|--------|-----|----------|-------|----------|-------|---------|-----------|------|
| w/ mask | 43.40 | 63.80 | 48.40 | 32.20 | 35.60 | 13.80 | 15.28 | 36.07 |
| w/o mask | 41.60 | 61.00 | 46.60 | 29.80 | 33.60 | 13.80 | 15.28 | 34.53 |

Table 7: Effect of Document Token Loss Masking. We compare model performance with and without loss masking applied to document tokens.

We observe that all three algorithms effectively enhance the model's ability to perform search within our framework, demonstrating the strong generalization ability of ZEROSEARCH. Among them, REINFORCE achieves the best performance, consistent with the observations in Search-R1 Jin et al. (2025), further highlighting its advantages in training stability. It is also worth noting that both REINFORCE and GRPO involve repeated rollouts, which incur higher API costs when interacting with a real search engine. This further underscores the practicality of our simulated search setup.

## D  EFFECT OF DOCUMENT TOKEN LOSS MASKING

During training, we apply loss masking to document tokens, as they are not generated by the policy model and may introduce noise. To assess the impact of the loss masking, we conduct ablation experiments using the LLaMA-3.2-3B model. The resulting reward curves are shown in Figure 4c.

As illustrated, removing document token loss masking leads to a substantial drop in the reward, indicating that including such tokens in the loss computation causes training instability and degrades learning effectiveness. Table 7 further supports this observation, showing a significant decline in model performance without loss masking.

## E  COST ANALYSIS

Although our method eliminates the cost associated with commercial APIs, it requires the deployment of simulated search LLMs on GPU servers. In this section, we compare the cost of our method with that of commercial search API-based approaches. API costs are estimated based on the pricing of SerpAPI[2], while GPU deployment costs are based on Amazon Web Services (AWS)[3]. In this paper, we perform reinforcement learning (RL) training with a batch size of 64, a rollout repetition of 5, and a total of 200 training steps. This setup results in approximately 12 hours of training when using Qwen-2.5-7B as the backbone, generating approximately 64,000 search requests.

As shown in Table 8, the cost of using Google Search is significantly higher than that of our approach. Furthermore, the API cost increases substantially with larger batch sizes or more training steps, making the method less scalable due to prohibitively high expenses. In contrast, ZEROSEARCH offers a more cost-efficient alternative. It is important to note that GPU utilization of the simulation server is inconsistent—high during the rollout and low during policy update—resulting in periods of underutilization and resource waste. A practical strategy to mitigate this issue is to share a single simulation server across multiple RL training tasks, which can further reduce deployment costs. In addition, we provide simulation LLMs of various parameter sizes, giving users flexible options to balance performance and resource usage based on their specific needs.

---

[2]https://serpapi.com/pricing
[3]https://aws.amazon.com/ec2/capacityblocks/pricing

| Search Engine | Queries | Training Time | GPUs Used | API Cost | GPU Cost | Total Cost |
|---|---|---|---|---|---|---|
| SFT-3B | ~64,000 | ~12 hours | $1 \times$ A100 GPUs | $0.0 | $17.7 | **$17.7** |
| SFT-7B | ~64,000 | ~12 hours | $2 \times$ A100 GPUs | $0.0 | $35.4 | **$35.4** |
| SFT-14B | ~64,000 | ~12 hours | $4 \times$ A100 GPUs | $0.0 | $70.8 | **$70.8** |
| Google | ~64,000 | ~12 hours | None | $586.7 | $0.0 | **$586.7** |

Table 8: Cost comparison between real search engine and our simulated search approach.

## F   CASE STUDY

In this section, we present case studies from the interaction trajectories and the simulated documents to further illustrate the effectiveness of our proposed method.

**Interaction Trajectory Study**   We first show several interaction trajectories in Table 9. From these examples, we make the following observations: First, the policy model consistently adheres to the expected output format, which is surprising given that the format requirements are only specified in the input template and are not explicitly reinforced through the reward design. Second, the model demonstrates the capability for multi-turn search behavior to arrive at the final answer. This is a crucial finding, as it confirms that our method effectively incentivizes the model's search capabilities.

**Simulated Document Study**   We further present examples of simulated document outputs generated by the simulation LLMs in Table 10. By comparing useful outputs with noisy ones, we observe a clear distinction in quality. Notably, the useful outputs consistently contain the correct answer, while the noisy outputs fail to do so. Precise control over document quality enables the implementation of a curriculum-based rollout mechanism and contributes to stabilizing the reinforcement learning training process.

## G   IMPLEMENTATION DETAILS

During experimentation, we deploy the simulation server on 4 H20 GPUs and conduct RL training on another 4 H20 GPUs. To train the simulation LLM, we conduct a lightweight SFT using Qwen-2.5-3B-Instruct, Qwen-2.5-7B-Instruct, and Qwen-2.5-14B-Instruct as the backbones. The learning rate is set to be 1e-6. To train ZEROSEARCH, we adopt three reinforcement learning algorithms: REINFORCE, GRPO, and PPO. For both REINFORCE and GRPO, the policy LLM is trained with a learning rate of 1e-6, and five responses are sampled per prompt. In the PPO setting, the policy LLM is trained with a learning rate of 1e-6, while the value model is trained with a separate learning rate of 1e-5. We apply Generalized Advantage Estimation (GAE) with hyperparameters $\lambda = 1$ and $\gamma = 1$. For Qwen-2.5-7B (Base/Instruct), the initial noise probability $p_s$ and final noise probability $p_e$ are set to 0 and 0.75. For Qwen-2.5-3B (Base/Instruct), they are set to 0 and 0.25. For LLaMA-3.2-3B (Base/Instruct), they are set to 0.25 and 0.5. Unless otherwise specified, all RL-based methods use REINFORCE as the default training algorithm. The fine-tuned Qwen-2.5-14B-Instruct model serves as the default simulated search engine across all experiments. During inference, we uniformly use Google Web Search as the search engine for all methods.

**Question**: What is the place of birth of Clara Novello's father?
**Answer**: London, England
ZEROSEARCH:
`<think>` The question is about the place of birth of Clara Novello's father. `</think>`
`<search>` place of birth of Clara Novello's father `</search>`
`<information>` Doc 1: 68 Clara Novello's Reminiscences In this interesting small place I knew the daughter-in-law of Goethe, who showed us his house, his bedroom, the gold laurel Doc 2: So, we may happily add James Clerk Maxwell to our list as this year is the 175th since his birth in Edinburgh on November 13th, 1831. Doc 3: When I was intro- duced to him as Vincent Novello's eldest daughter, Coleridge was struck by my father's name, knowing it to be that of a musician, and Doc 4: ABSTRACT. This publication presents biographical profiles of people of interest to young readers. The concept is unique in that. Doc 5: Reuse: Copies of full items can be used for personal research or study, educational, or not-for-profit purposes without prior permission or charge. `</information>`
`<think>` Based on the search results, the place of birth of Clara Novello's father is not directly mentioned. However, I can infer that Clara Novello's father is Vincent Novello, and he is a musician. Since the search results do not provide the place of birth of Vincent Novello, I need to search for his place of birth. `</think>`
`<search>` place of birth of Vincent Novello `</search>`
`<information>` Doc 1: Born: September 6, 1781, London, England. Died: August 9, 1861, Nice, France. Buried: Cimetiére du Château, Nice, France. portrait. Biography. Doc 2: NOVELLO, VINCENT (1781–1861), organist, musical composer, editor, and arranger, was born at 240 Oxford Road (now Oxford Street), London, Doc 3: Vincent Novello Leigh Hunt was born about 1823 in Genoa, Italy, where his parents, writer Leigh Hunt and his wife Marianne Kent, had been living Doc 4: NOVELLO, Vincent, son of an Italian father and English mother, was born at 240, Oxford Street, Sept. 6, 1781. Doc 5: Vincent Novello (6 September 1781 – 9 August 1861), was an English musician and music publisher born in London. `</information>`
`<think>` Based on the search results, the place of birth of Vincent Novello, Clara Novello's father, is London, England. `</think>`
`<answer>` London, England `</answer>`

**Question**: Who is the spouse of the person who played the sergeant major in We Were Soldiers?
**Answer**: Katharine Ross
ZEROSEARCH:
`<think>` The question is about the spouse of the person who played the sergeant major in the movie "We Were Soldiers". `</think>`
`<search>` who played the sergeant major in We Were Soldiers `</search>`
`<information>` Doc 1: Maj. Basil L. Plumley. Sam Elliott was the only actor who Randall Wallace could envision as the battalion's hard-as-nails sergeant major. Doc 2: But apparently, Plumley had no issues with Elliott's portrayal. The actor "under played" the sergeant major, said Galloway, who introduced the Doc 3: The U.S. Army is investigating allegations that the late Command Sgt. Major Basil Plumley wore unauthorized combat and valor awards. Doc 4: Plumley was played by actor Sam Elliott in the movie We Were Soldiers (2002) about the Battle of Ia Drang. Command Sergeant Major Basil L. Doc 5: I'll start by saying Plumley was a badass (edit: apparently not, TIL), and that Sam Elliott's portrayal was equally brilliant. `</information>`
`<think>` Based on the search results, the person who played the sergeant major in "We Were Soldiers" is Sam Elliott. Now I need to find out who his spouse is. `</think>`
`<search>` spouse of Sam Elliott `</search>`
`<information>` Doc 1: Sam Elliott has been married to Katharine Ross since 1984, indicating a long-term heterosexual relationship. There is no credible evidence to Doc 2: Actor Sam Elliott, star of the film "The Hero," talks Hollywood, his 30-year marriage to actress Katherine Ross and his famous mustache. Doc 3: Sam Elliott and Katharine Ross met in 1978 and were married in 1984. They met while filming the horror movie, The Legacy. It was Ross's Doc 4: After more than 35 years of marriage, it's clear that Sam Elliott and his wife, Katharine Ross, are one of Hollywood's most successful love Doc 5: Sam Elliott and his wife Katharine Ross have been married for nearly four decades. Their love story dates back to the late 1960s. `</information>`
`<think>` Based on the search results, the spouse of Sam Elliott, who played the sergeant major in "We Were Soldiers," is Katharine Ross. `</think>`
`<answer>` Katharine Ross `</answer>`

Table 9: Interaction trajectories of ZEROSEARCH. The results demonstrate that the policy model learns to output the correct format and effectively conducts multi-step search to reach the final answer.

**Question**: who came second in tour de france 2018?
**Answer**: Tom Dumoulin
**Query**: Tour de France 2018 second place

Useful Output:
Doc 1: The 2018 Tour de France was the 105th edition of the race, and the first under the leadership of new event president Étienne Thobois.
Doc 2: 2nd – Tom Dumoulin (NED / Team Sunweb) 3rd – Geraint Thomas (GBR / Team Sky) 4th – Jakob Fuglsang (DEN / Astana) 5th – Romain Bardet (FRA / AG2R
Doc 3: 2018 Tour de France – Stage 20. 15:20 | 51:58. Stage 20 is the final time trial of this year's Tour de France. The 28.6km parcours will start and finish
Doc 4: Tour de France 2018: Live coverage of final stage | Cycling News · Stage 21 live commentary. 1.1M views · 1 year ago #cyclingnews · Tour de France 2018: Stage 21 -
Doc 5: 2018 TDF. Egan Bernal. 29. Chris Froome. 30. Tom Dumoulin. 31. Geraint Thomas. 32. Romain Bardet. 33. Simon Yates. 34. Thibaut Pinot. 35.

Noisy Output:
Doc 1: second place is 5000 euros, with 3000 euros for the third-placed rider. Riders in the top three of each stage receive a cash bonus as well.
Doc 2: second place was 1 minute and 47 seconds. This time gap is the biggest ever in the history of Tour de France. The previous record is 1 minute
Doc 3: 1978. Second place, Joop Zoetemelk. 1980. Not Having Enough Time. An hour before Wouter Weylandt won stage nine, a picture of the Tour 2020 got painted.
Doc 4: second place in January 2017. Over the following two years, the Swiss rider won the Giro d'Italia and Tour de France, adding the Vuelta to Spain in
Doc 5: The awards for best riders in their respective categories were presented at the end of each stage and the overall rankings are published every evening.

**Question**: What is the name of the sequel to the novel "The Alienist" by author Caleb Carr?
**Answer**: The Angel of Darkness
**Query**: sequel to The Alienist Caleb Carr

Useful Output:
Doc 1: Caleb Carr. THE ALIENIST. A Novel. Vintage Books Random House New York. Contents. Acknowledgments. Warning 7. Epigraph. 13. Prologue. 15. 1896. 18. 1. 21.
Doc 2: Book review: The Alienist by Caleb Carr · Comments5. The Alienist by Caleb Carr. Rating: 5 out of 5. 1896 – 1897. New York City, 1896 is a time
Doc 3: The Alienist: A Novel (2017) · The Angel of Darkness (2018) · The Lost City of Z (2019) · The Devil in the White City (2019) · A Gentleman in Moscow (2019)
Doc 4: The sequel to the acclaimed national bestseller The Alienist, Caleb Carr's The Angel of Darkness is a breathtaking thriller set in 1897 New York.
Doc 5: The Angel of Darkness is a 1994 horror detective novel by American author Caleb Carr. It is the sequel to Carr's best-selling debut novel

Noisy Output:
Doc 1: The Alienist has been adapted for the screen, the eight-part miniseries coming to production this summer and airing in January 2018 on TNT.
Doc 2: The Alienist is a 1994 American suspense novel by Caleb Carr. Set in New York City during the winter of 1896, it follows the investigation of a
Doc 3: Imagine an alienist, an alienist is a psychiatrist in the 19th century. Okay, so who's the alienist? John Corrigan. John Corrigan played by Daniel
Doc 4: Revisit Caleb Carr's The Alienist by watching the classic crime drama, now streaming on HBO Max.
Doc 5: Join us this week as we explore Caleb Carr's fictional world of monsters and monsters hunters in our conversation with Caleb about his most

Table 10: Outputs from the 14B simulation LLMs. The correct answers are highlighted in red. We can find that the quality of the useful output is much better than that of the noisy output.