# OpenReview forum: "ZeroSearch: Incentivize the Search Capability of LLMs without Searching"
_ICLR.cc/2026/Conference — ICLR 2026 Conference Desk Rejected Submission_

### Official Review · Reviewer_pjfW · 2025-10-28

**Soundness:** 1
**Presentation:** 3
**Contribution:** 1
**Rating:** 2
**Confidence:** 5

**Summary:**

This paper proposes ZeroSearch, an RL framework that trains LLMs to develop search capabilities without interacting with a real search engine. The method replaces the real search environment (e.g., Google via SerpAPI) with a simulated search LLM, which is fine-tuned to generate both “useful” and “noisy” documents. During RL training, a curriculum rollout progressively increases noise levels to encourage robustness. Experiments shows improved Exact Match (EM) performance with zero API cost.

**Strengths:**

* Addresses an important practical bottleneck in RL-based search training — high API costs and unstable document quality — with a controlled simulation-based setup.
* The curriculum degradation and document quality control mechanisms are conceptually reasonable and could improve stability in RL training.
* The writing and experimental organization are clear and the comparisons cover multiple model families (Qwen, LLaMA).

**Weaknesses:**

1. **Metric validity (critical)**

The paper exclusively reports Exact Match (EM) as its main evaluation metric. However, EM doesn't capture search quality,  **especially for prompting-based method** (an LLM fine-tuned w/ EM could outperform prompting-based method w/ search), so the performance table is not trustworthy.

2. **Unfair baseline setup**

For non-multi-turn baselines like RAG, the comparison is imbalanced. If ZeroSearch effectively accesses 4 turns × 3 documents = 12 documents (at max.), then RAG should also be given the top 12 documents (from one-shot retrieval) for a fair comparison. Otherwise, the improvement might come from context size rather than genuine search ability.

3. **Missing key baseline (RAG + SFT)**

The paper omits a critical baseline — RAG + SFT (see *"Baselines with retrievals."* in Self-RAG [1]), i.e., fine-tuning on the top retrieved documents with the ground-truth answer as supervision. This baseline is widely recognized and provides a fair supervised comparison without RL. Without it, the claimed gains from ZeroSearch cannot be meaningfully attributed to the proposed RL method.

4. **Limited novelty**

The main “innovation” — using a fine-tuned LLM to simulate a search engine — is conceptually incremental. Prior works already train RL agents on retrieval tasks; substituting the search engine with a simulator mainly reduces cost, rather than introducing a new algorithmic principle. The contribution feels more like engineering optimization than a conceptual breakthrough.

5. **Training inefficiency**

As noted by [2, 3], training uniformly on all examples (easy and hard) is inefficient. ZeroSearch’s curriculum schedule is fixed and lacks adaptive difficulty adjustment, potentially wasting computation and limiting scalability.


[1] Asai et al., Self-RAG: Learning to Retrieve, Generate, and Critique through Self-Reflection., ICLR 2024.

[2] Jiang et al., s3: You Don’t Need That Much Data to Train a Search Agent via RL., EMNLP 2025.

[3] Mei, et al., "AI-SearchPlanner: Modular Agentic Search via Pareto-Optimal Multi-Objective Reinforcement Learning." arXiv 2025.08.

**Questions:**

1. Was Search-R1 trained using SerpAPI (Google Search) or a static Wikipedia corpus? Section 4.3 states all evaluations use SerpAPI, but it’s **unclear for training**. If Search-R1 was trained on Wikipedia, the comparison is unfair because the domain and style differ significantly.

2. How sensitive are the results to the number of documents per turn or the curriculum parameters (e.g., noise probability schedule)?

---

> ### Author Response · Authors · 2025-11-17
> **Response to Reviewer pjfW (Part 1)**
>
> Thank you very much for the detailed feedback. We address your concerns point by point below.
>
> ### **1. EM doesn't capture search quality,**
>
> Thank you for raising this important concern. While EM is a standard metric in the RAG literature, we agree that it may not fully reflect search quality. To address this, we conducted an additional evaluation using the **F1 score**, which provides a more nuanced measure of answer overlap. Results on the Qwen2.5-7B series are shown below:
>
> | Method          |  NQ   | TriviaQA | PopQA | HotpotQA | 2WikiMQA | MuSiQue | Bamboogle |  Avg  |
> | :-------------- | :---: | :------: | :---: | :------: | :------: | :-----: | :-------: | :---: |
> | Direct          | 17.48 |  43.47   | 6.94  |  21.49   |  27.83   |  11.08  |   23.58   | 21.70 |
> | CoT             | 19.80 |  44.38   | 9.90  |  20.06   |  26.28   |  14.37  |   30.55   | 23.62 |
> | RAG             | 38.51 |  67.32   | 30.43 |  34.35   |  29.13   |  17.88  |   26.85   | 34.92 |
> | RAG-Agent       | 28.62 |  46.70   | 13.07 |  26.70   |  26.70   |  16.30  |   33.00   | 27.30 |
> | Search-O1       | 25.77 |  48.36   | 16.55 |  24.98   |  31.39   |  16.96  |   40.55   | 29.22 |
> | R1-base         | 29.28 |  49.34   | 18.19 |  27.40   |  34.05   |  18.62  |   32.58   | 29.92 |
> | R1-instruct     | 29.26 |  48.60   | 19.83 |  28.81   |  32.40   |  16.64  |   33.65   | 29.88 |
> | Search-R1-base  | 45.50 |  65.16   | 32.41 |  36.32   |  36.71   |  24.95  |   37.78   | 39.83 |
> | Search-R1-inst  | 44.21 |  67.35   | 35.28 |  40.30   |  36.52   |  24.85  |   33.81   | 40.33 |
> | ZeroSearch-base | 43.29 |  67.78   | 33.08 |  38.13   |  37.57   |  24.23  |   39.41   | 40.50 |
> | ZeroSearch-inst | 46.02 |  69.33   | 33.69 |  41.77   |  38.46   |  25.31  |   38.33   | 41.84 |
>
> ZeroSearch matches or surpasses Search-R1 across tasks, confirming that our improvements hold even under a more robust metric such as F1.

---

> ### Author Response · Authors · 2025-11-17
> **Response to Reviewer pjfW (Part 2)**
>
> ### **2. Unfair baseline setup and Missing RAG+SFT baseline**
>
> Thank you for this suggestion. We have added two critical baselines: RAG-SFT and RAG with 10 documents. Results across three model families are presented below:
>
> **Qwen2.5-7B-Instruct**
>
> | Method          |  NQ   | TriviaQA | PopQA | HotpotQA | 2WikiMQA | MuSiQue | Bamboogle |  Avg  |
> | :-------------- | :---: | :------: | :---: | :------: | :------: | :-----: | :-------: | :---: |
> | Direct Answer   | 11.60 |  35.60   | 1.20  |  16.40   |  22.20   |  4.80   |   14.40   | 15.17 |
> | CoT             | 12.80 |  35.60   | 3.80  |  16.20   |  22.60   |  6.60   |   24.00   | 17.37 |
> | RAG (5 doc)     | 27.40 |  58.20   | 17.80 |  25.80   |  23.20   |  9.40   |   16.80   | 25.51 |
> | RAG (10 doc)    | 27.40 |  59.40   | 17.40 |  27.20   |  24.22   |  11.00  |   19.20   | 26.55 |
> | RAG-SFT         | 21.40 |  53.20   | 17.00 |  23.60   |  23.81   |  10.00  |   16.80   | 23.69 |
> | RA-Agent        | 21.20 |  40.20   | 8.80  |  19.60   |  19.60   |  7.60   |   28.00   | 20.71 |
> | Search-o1       | 19.40 |  40.60   | 11.40 |  17.00   |  27.00   |  8.60   |   30.40   | 22.06 |
> | R1-base         | 27.60 |  47.40   | 27.40 |  21.00   |  29.20   |  9.80   |   27.78   | 27.17 |
> | R1-instruct     | 27.00 |  45.80   | 24.20 |  21.60   |  27.80   |  8.40   |   25.00   | 25.69 |
> | Search-R1-base  | 43.40 |  61.40   | 54.60 |  31.20   |  37.20   |  18.20  |   30.56   | 39.51 |
> | Search-R1-inst  | 42.40 |  63.40   | 51.60 |  32.80   |  33.20   |  17.40  |   26.39   | 38.17 |
> | ZeroSearch-base | 42.40 |  66.40   | 60.40 |  32.00   |  34.00   |  18.00  |   33.33   | 40.93 |
> | ZeroSearch-inst | 43.60 |  65.20   | 48.80 |  34.60   |  35.20   |  18.40  |   27.78   | 39.08 |
>
> **Qwen2.5-3B-Instruct**
>
> | Method          |  NQ   | TriviaQA | PopQA | HotpotQA | 2WikiMQA | MuSiQue | Bamboogle |  Avg  |
> | :-------------- | :---: | :------: | :---: | :------: | :------: | :-----: | :-------: | :---: |
> | Direct Answer   | 12.40 |  30.60   | 5.60  |  16.00   |  19.20   |  4.40   |   16.80   | 15.00 |
> | CoT             | 15.00 |  33.60   | 3.60  |  16.20   |  18.00   |  3.60   |   12.80   | 14.69 |
> | RAG (5 doc)     | 31.60 |  58.00   | 15.20 |  24.20   |  23.20   |  8.20   |   15.20   | 25.09 |
> | RAG (10 doc)    | 32.40 |  61.60   | 14.00 |  24.20   |  26.06   |  9.00   |   17.60   | 26.41 |
> | RAG-SFT         | 32.00 |  57.60   | 11.20 |  27.61   |  30.34   |  11.41  |   21.60   | 27.39 |
> | RA-Agent        | 15.20 |  28.40   | 6.60  |  12.60   |  16.60   |  2.60   |   13.60   | 13.66 |
> | Search-o1       | 16.60 |  31.00   | 8.20  |  14.80   |  22.40   |  5.20   |   22.40   | 17.23 |
> | R1-base         | 14.20 |  34.80   | 20.80 |  19.60   |  28.40   |  6.40   |   5.56    | 18.54 |
> | R1-instruct     | 19.80 |  33.00   | 19.40 |  19.40   |  26.40   |  4.40   |   11.11   | 19.07 |
> | Search-R1-base  | 40.60 |  60.00   | 44.20 |  29.20   |  32.00   |  11.20  |   12.50   | 32.81 |
> | Search-R1-inst  | 35.80 |  55.80   | 26.00 |  33.20   |  26.00   |  7.60   |   12.50   | 28.13 |
> | ZeroSearch-base | 43.00 |  61.60   | 41.40 |  33.80   |  34.60   |  13.00  |   13.89   | 34.47 |
> | ZeroSearch-inst | 41.40 |  57.40   | 44.80 |  27.40   |  30.00   |  9.80   |   11.11   | 31.70 |
>
> **Llama-3.2-3B-Instruct**
>
> | Method          |  NQ   | TriviaQA | PopQA | HotpotQA | 2WikiMQA | MuSiQue | Bamboogle |  Avg  |
> | :-------------- | :---: | :------: | :---: | :------: | :------: | :-----: | :-------: | :---: |
> | Direct Answer   | 16.20 |  29.60   | 7.40  |  12.60   |   9.20   |  2.00   |   8.00    | 12.14 |
> | CoT             | 26.20 |  44.40   | 2.80  |  16.00   |  10.20   |  5.80   |   21.60   | 18.14 |
> | RAG (5 doc)     | 30.00 |  57.60   | 26.40 |  23.40   |  17.60   |  9.60   |   11.20   | 25.11 |
> | RAG (10 doc)    | 29.80 |  61.00   | 27.00 |  26.61   |  19.64   |  9.20   |   15.20   | 26.92 |
> | RAG-SFT         | 30.80 |  60.60   | 17.60 |  26.61   |  19.64   |  14.21  |   24.00   | 27.64 |
> | RA-Agent        | 22.40 |  36.20   | 11.40 |  16.60   |  21.00   |  5.60   |   26.40   | 19.94 |
> | Search-o1       | 24.20 |  48.40   | 8.80  |  19.40   |  17.40   |  6.00   |   32.00   | 22.31 |
> | R1-base         | 28.40 |  44.20   | 30.00 |  22.80   |  28.40   |  7.00   |   11.11   | 24.56 |
> | R1-instruct     | 35.00 |  52.20   | 27.60 |  21.60   |  17.80   |  11.40  |   20.83   | 26.63 |
> | Search-R1-base  | 41.20 |  60.00   | 44.00 |  29.60   |  31.60   |  13.60  |   19.44   | 34.21 |
> | Search-R1-inst  | 37.60 |  53.60   | 44.20 |  21.00   |  20.40   |  8.80   |   27.78   | 30.48 |
> | ZeroSearch-base | 43.40 |  63.80   | 48.40 |  32.20   |  35.60   |  13.80  |   15.28   | 36.07 |
> | ZeroSearch-inst | 40.20 |  58.00   | 46.00 |  22.80   |  21.40   |  10.40  |   18.06   | 30.98 |

---

> > ### Author Response · Authors · 2025-11-17
> > **Response to Reviewer pjfW (Part 3)**
> >
> > Based on the result above, we can have the following observations:
> >
> > - While adding more documents (RAG 10 doc) provides marginal improvements over RAG (5 doc), the gains are limited. This is because Google's ranking already places the most relevant documents at the top; adding more documents introduces noise that can distract the model.
> > - The results reveal an interesting pattern across model scales. RAG-SFT achieves better performance than vanilla baselines on smaller models but underperforms on the larger Qwen2.5-7B model. This is primarily because larger models have stronger inherent capabilities for information extraction and reasoning, making the benefits of SFT less pronounced, while smaller models benefit more significantly from fine-tuning on retrieval-augmented examples.
> >
> > Overall, these baselines validate that ZeroSearch’s gains do not come from document volume or SFT, but from its multi-step, adaptive RL training.
> >
> > ### **3. Limited novelty**
> >
> > We respectfully disagree with this assessment. Our contributions are:
> >
> > - **First simulated search environment for agentic RL**: We introduce a novel paradigm that eliminates API dependencies while maintaining training effectiveness—a contribution also acknowledged by Reviewer vjRe.
> > - **Curriculum-based noise injection**: Our progressive difficulty adjustment mechanism significantly improves training stability and final performance, which is absent in prior work.
> > - **Practical impact**: We achieve superior performance at **zero API cost**, making RL-based search training accessible to the broader research community.
> >
> > We believe our work opens a new direction: **learning not only the policy, but also the environment**. As LLMs continue to improve, their ability to simulate realistic retrieval environments will also grow, making this paradigm more powerful and widely applicable in the future.
> >
> > ### **4. Training inefficiency**
> >
> > We believe there is a misunderstanding regarding our curriculum design. Our curriculum schedule is defined by the probability function:
> > $$
> > p_i = p_s + \frac{b^{i/m - 1}}{b - 1}(p_e - p_s)
> > $$
> > where $p_i $ is the noise probability at training step $i $, $m $ is the total number of training steps, $p_s $ and $p_e $ are the start and end probabilities, and $b $ controls the curve shape.
> >
> > - The parameters $p_s $ and $p_e $ allow easy adjustment of training difficulty, providing flexible control over the curriculum progression.
> > - While our current implementation uses training progress (step $i $) as the controller, our framework supports alternative difficulty metrics such as pass rate, validation performance, or task-specific indicators, which could provide more tailed control based on the model's status.
> >
> > ### **5. Search-R1 training setup**
> >
> > Sorry for the confusion. Search-R1 was trained using SerpAPI (Google Search), consistent with our evaluation setup. The comparison is therefore fair and conducted under identical conditions. We will clarify this in the next verison of our paper to avoid confusion.
> >
> > ### **6. Sensitivity to hyperparameters**
> >
> > We appreciate this question and hope to clarify as follows:
> >
> > **Documents per turn:** Our experiments demonstrate that the number of documents per turn has limited impact once a reasonable threshold is met (as shown in the RAG 5 doc vs. 10 doc comparison above). The search engine's ranking already prioritizes relevant documents, so additional documents mainly introduce noise.
> >
> > **Curriculum parameters:** Our ablation studies show that: stronger models benefit from steeper difficulty curves (larger $p_e - p_s $), while weaker models require gentler progression (smaller $p_e - p_s $). This flexibility allows our framework to adapt to different model capacities.
> >
> > We hope these clarifications address your concerns. We are happy to provide additional experiments or clarifications as needed.

---

> ### Comment · Reviewer_pjfW · 2025-11-17
>
> Thanks for your response. Could you provide the script for the RAG+SFT baseline? I would like to run it myself.
>
> You can provide all the data for training, including top-10 retrieval results. (Note: those top-10 retrieved docs for training should be from Serp but not be synthesized ones), and it is better to also provide the inference script for all the single-turn retrieval baselines.

---

> ### Comment · Reviewer_pjfW · 2025-11-17
>
> For your other responded items:
>
> "First simulated search environment for agentic RL": It's not something more novel than synthetic data. The motivation is to reduce the training cost, but the heavy training cost was caused by using very sparse rewards like EM, not something else.
>
> The comparison between
>
> (a) [creating training data for tuning a "search engine" LLM with 14B model] + [training a search agent w/ RL + EM]
>
> and (b) [training a search agent w/ RL + real search engine + a more effective reward]
>
> is missing, but directly determines the meaningfulness of training such a simulated "env".

---

> > ### Author Response · Authors · 2025-11-30
> >
> > Thank you for your follow-up questions! We are happy to provide the requested details.
> >
> > **1. RAG+SFT Training and Inference Setup**
> >
> > We use [LlamaFactory](https://github.com/hiyouga/LLaMA-Factory) for training the RAG+SFT baseline. The training data consists of approximately 20k question-answer pairs augmented with top-10 documents retrieved from SerpAPI (Google Search). For inference, we use the following prompt template:
> >
> > ```
> > Use only the information from the following documents to answer the question with one short phrase.
> > Documents: {retrieved_docs}
> > Question: {question}
> > Output:
> > ```
> >
> > This prompt is designed to encourage concise, extractive answers based solely on the retrieved content.
> >
> > **2. Regarding Your Other Comments**
> >
> > We appreciate your perspective, but would like to clarify:
> >
> > - **On novelty**: While synthetic data generation is not new, our contribution lies in using an LLM to simulate an interactive search environment for RL training—not merely generating static training examples. Furthermore, we introduce a curriculum-based rollout strategy that progressively increases document noise during training, which significantly improves training stability and final performance. This combination of simulated environment and curriculum learning is novel in the context of search-augmented LLMs.
> >
> > - **On reward design**: We acknowledge that denser rewards could improve training efficiency. However, our work focuses on a different axis: *environment simulation*. The comparison you suggest—RL with real search + better rewards vs. RL with simulated search + EM—is orthogonal to our contribution. Our simulated environment is compatible with any reward function, and we expect combining ZeroSearch with improved rewards would yield further gains.
> >
> > We will prepare and release the complete training scripts, data, and inference code to facilitate reproducibility. Thank you again for the constructive discussion.

---

### Official Review · Reviewer_ckY5 · 2025-11-01

**Soundness:** 3
**Presentation:** 3
**Contribution:** 3
**Rating:** 8
**Confidence:** 3

**Summary:**

The paper proposes **ZeroSearch**, a RL framework designed to equip LLMs with the ability to use search engines when their internal knowledge is insufficient. The authors identify two main challenges with prior search-augmented RL setups: 1) the high cost of real API calls and 2) the uncontrolled quality of retrieved documents, and address these by introducing a simulated retrieval module. This module is another LLM, fine-tuned to act as a noisy search engine, generating both relevant and irrelevant documents to mimic real search conditions. During training, the policy model interacts with this simulator through a curriculum rollout strategy that gradually increases retrieval noise, allowing the model to learn effective search behaviors in a controlled environment. Experiments across multiple QA benchmarks and model families show that ZeroSearch matches the performance of models trained with real search engines when using a 7B retrieval simulator, and surpasses them when scaling to 14B.

**Strengths:**

- The paper is well written, and the proposed approach is clearly presented.
- The main idea of the framework is interesting and well motivated. Since the goal of RL training here is to teach the model how to use a search tool, simulating search with a fixed, noisy model that provides outdated or imperfect information effectively avoids the cost of real API calls while improving robustness to inconsistent or low-quality retrievals.
- The experimental section is extensive and promising, covering a diverse set of baselines, and datasets. The results are consistently strong and demonstrate both the strength and scalability of the approach.
- The cost analysis of Appendix E is informative, clearly demonstrating the efficiency of ZeroSearch. The comparison shows that fine-tuning the retrieval module is substantially less expensive than performing repeated real search API calls during RL training.

**Weaknesses:**

- The paper would benefit from quantitative results on the robustness of the reward choice. The authors mention that exact matching was prone to reward hacking, but this claim is only stated qualitatively. Including some quantitative results, or an additional ablation study comparing reward formulations, would make the argument stronger and more convincing.

**Questions:**

- Why did the authors directly chose a RL approach over a SFT strategy, as used in works like Toolformer [1]? Given that the search simulating model is already trained, it should be relatively inexpensive to generate large amounts of multi-turn synthetic data without relying on real APIs. Would performing SFT on such self-generated interactions also yield a strong model, while avoiding the need to design nuanced reward functions?

[1] Schick, Timo, et al. "Toolformer: Language models can teach themselves to use tools." Advances in Neural Information Processing Systems 36 (2023): 68539-68551.

---

> ### Author Response · Authors · 2025-11-17
>
> Thank you very much for the thoughtful review and positive assessment. We address your question below.
>
> ### **1. On the Choice of Reward Function**
>
> Thank you for pointing this out. In our initial experiments, we used exact match (EM) as the reward, but we observed that training quickly collapsed due to reward hacking: the model learned to output entire retrieved documents to guarantee that the ground-truth answer appeared as a substring. Since EM only checks for containment, this behavior was rewarded despite being clearly undesirable.
>
> We therefore switched to the F1 score, which measures token-level overlap between the prediction and the gold answer. This discourages degenerate behaviors and results in much more stable learning. We will clarify this in the next revision.
>
> ### **2. Why use RL instead of an SFT-style approach like Toolformer?**
>
> Thank you for the insightful question. We choose RL because search is an *interactive, multi-step* process. Each query depends on the retrieved documents from previous steps, and the model often needs to repair suboptimal queries or explore alternative search strategies. RL trains the model on its *own* trajectories, allowing it to learn this adaptive behavior naturally.
>
> In contrast, SFT is fundamentally off-policy: it can only imitate fixed traces and cannot teach the model how to recover from errors or handle noisy retrieval dynamically. This limitation becomes especially significant for multi-hop QA, where iterative refinement is essential.
>
> ### **3. Would SFT on simulator-generated multi-turn data be sufficient?**
>
> Thank you for this excellent suggestion. To verify this, we conducted additional experiments where we used the simulation LLM to generate large amounts of search-augmented traces and performed pure SFT (“Simulate SFT”). Below are the results on Llama-3.2-3B-Instruct:
>
> | Method           |  NQ   | TriviaQA | PopQA | HotpotQA | 2WikiMQA | MuSiQue | Bamboogle |  Avg  |
> | :--------------- | :---: | :------: | :---: | :------: | :------: | :-----: | :-------: | :---: |
> | Direct Answer    | 16.20 |  29.60   | 7.40  |  12.60   |   9.20   |  2.00   |   8.00    | 12.14 |
> | CoT              | 26.20 |  44.40   | 2.80  |  16.00   |  10.20   |  5.80   |   21.60   | 18.14 |
> | RAG              | 30.00 |  57.60   | 26.40 |  23.40   |  17.60   |  9.60   |   11.20   | 25.11 |
> | RA-Agent         | 22.40 |  36.20   | 11.40 |  16.60   |  21.00   |  5.60   |   26.40   | 19.94 |
> | Search-o1        | 24.20 |  48.40   | 8.80  |  19.40   |  17.40   |  6.00   |   32.00   | 22.31 |
> | R1-base          | 28.40 |  44.20   | 30.00 |  22.80   |  28.40   |  7.00   |   11.11   | 24.56 |
> | R1-instruct      | 35.00 |  52.20   | 27.60 |  21.60   |  17.80   |  11.40  |   20.83   | 26.63 |
> | **Simulate SFT** | 33.40 |  64.00   | 16.80 |  28.21   |  21.23   |  14.21  |   25.20   | 29.01 |
> | Search-R1-base   | 41.20 |  60.00   | 44.00 |  29.60   |  31.60   |  13.60  |   19.44   | 34.21 |
> | Search-R1-inst   | 37.60 |  53.60   | 44.20 |  21.00   |  20.40   |  8.80   |   27.78   | 30.48 |
> | ZeroSearch-base  | 43.40 |  63.80   | 48.40 |  32.20   |  35.60   |  13.80  |   15.28   | 36.07 |
> | ZeroSearch-inst  | 40.20 |  58.00   | 46.00 |  22.80   |  21.40   |  10.40  |   18.06   | 30.98 |
>
> The results show that training with simulator-generated SFT data does provide noticeable improvements over vanilla RAG. However, across all architectures and model sizes, this SFT approach still falls short of ZeroSearch. This empirical evidence strengthens our conclusion that RL remains necessary for learning robust search behavior.
>
> We thank the reviewer again for the insightful feedback and constructive suggestions. They have helped us significantly improve the clarity and completeness of our work.

---

### Official Review · Reviewer_S5aY · 2025-11-01

**Soundness:** 2
**Presentation:** 3
**Contribution:** 2
**Rating:** 4
**Confidence:** 4

**Summary:**

This paper introduces ZeroSearch, a reinforcement learning (RL) framework for training LLMs to use search tools without using with a real search engine. This is motivated by two major challenges in calling searching engines: (1) unpredictable document quality and (2) high API costs from frequent search calls in RL rollouts. ZeroSearch works by first fine-tuning an LLM to act as a simulated retrieval module, which learns to generate both useful and noisy documents in response to a query. During training, it uses a curriculum-based strategy, starting with helpful documents and gradually increasing noisy contexts. Therefore, ZeroSearch progressively challenges the policy model and improve the models searching / context understanding. Experiments show ZeroSearch can reduce API costs and and achieve similar performance to models trained with a real search engine.

**Strengths:**

1. The proposed ZeroSearch trains the model using a simulated search engine, which removes the high costs of making frequent search calls during RL training.

2. Despite not using a real search engine, it solves the problem by using a simulating LLM and control the quality of generated documents with a curriculum learning-based rollout mechanism.

3. The proposed method is effective on single and multi-hop QA tasks, with experiments showing that simulator search engines can match the performance of a real search engine.

**Weaknesses:**

1. The training of ZeroSearch still requires extensive engine calls, which is used to curate the training data for the search simulation LLM. Consider that Search-R1 can converge within a few hundred steps, it is questionable if ZeroSearch really saves on the search costs.

2. While ZeroSearch can reduce API costs to train the policy model, the method introduces a new, significant computational cost in training and deploying a separate LLM as the simulation server, and the costs of which should be discussed in more details.

**Questions:**

1. How much engine calls are required to curate the training data for the search simulation LLM? How does it compares to the Search-R1 training paradigm with actual seach engine / embedding model retrieval?

---

> ### Author Response · Authors · 2025-11-17
>
> We sincerely thank the reviewer for the constructive feedback. We address the concerns below.
>
> ### **1. How many engine calls are required to curate the training data for the simulated search LLM?**
>
> Thank you for highlighting this important point.
>
> As noted, directly prompting an LLM to generate documents leads to a substantial style mismatch between simulated search outputs and real search results. To close this gap, we perform *simulation tuning*, where the LLM is fine-tuned on real query–document pairs.
>
> We curate **27.4k query–document pairs**, requiring **27.4k search engine calls**. This is a **one-time** data collection cost. Once the simulated search LLM is trained, it can be reused across tasks, datasets, and RL experiments *without requiring any additional API calls*. This makes the amortized cost substantially lower than approaches that repeatedly invoke real search APIs during RL rollouts.
>
> ### **2. Cost comparison with Search-R1 and other real-search–based paradigms**
>
> We appreciate the reviewer’s concern regarding computational overhead. To provide a clear comparison, we estimate both **training** and **deployment** (i.e., RL rollout) costs under standard pricing: SerpAPI pricing for web search, and AWS A100 capacity-block pricing for GPU usage.
>
> #### **One-time Training Cost**
>
> |    Component    | GPU Time |   GPUs   | Queries | GPU Cost | API Cost | Total Cost |
> | :-------------: | :------: | :------: | :-----: | :------: | :------: | :--------: |
> |  Simulator 3B   |    2h    | 4 × A100 |    —    |  $11.8   |    —     |   $11.8    |
> |  Simulator 7B   |    1h    | 4 × A100 |    —    |   $5.9   |    —     |    $5.9    |
> |  Simulator 14B  |   0.5h   | 4 × A100 |    —    |   $3.0   |    —     |    $3.0    |
> | Data Collection |    —     |    —     |  27.4k  |    —     |  $251.2  |   $251.2   |
>
> The dominant cost is the one-time collection of 27.4k real search queries. After this, the simulator can be reused indefinitely.
>
> #### **Per-experiment Deployment Cost (RL Rollouts)**
>
> (For this paper, we use batch size 64, rollout repeat 5, 200 steps, resulting in ~64k search queries.)
>
> |       Method       | Training Time |   GPUs   | Queries | GPU Cost | API Cost | Total Cost |
> | :----------------: | :-----------: | :------: | :-----: | :------: | :------: | :--------: |
> |  SFT-3B Simulator  |     ~12h      | 1 × A100 |  ~64k   |  $17.7   |   $0.0   | **$17.7**  |
> |  SFT-7B Simulator  |     ~12h      | 2 × A100 |  ~64k   |  $35.4   |   $0.0   | **$35.4**  |
> | SFT-14B Simulator  |     ~12h      | 4 × A100 |  ~64k   |  $70.8   |   $0.0   | **$70.8**  |
> | Search-R1 (Google) |     ~12h      |    —     |  ~64k   |   $0.0   |  $586.7  | **$586.7** |
>
> As shown above:
>
> - **Real-search-based RL (Search-R1)** is **8–30× more expensive** than ZeroSearch on a per-experiment basis.
> - The cost disparity grows **linearly with rollout steps and batch size**, making real-search-based RL increasingly impractical at scale.
>
> Finally, we emphasize that our experiments use a relatively small RL scale for demonstration. In industrial RL pipelines—where batch sizes, rollout lengths, and training hours are much larger—the API costs of real search would grow *exponentially*, while ZeroSearch remains stable.
>
> We thank the reviewer again for the thoughtful questions. We hope these clarifications help illustrate why ZeroSearch provides substantial long-term savings while maintaining competitive performance.

---

### Official Review · Reviewer_vjRe · 2025-11-04

**Soundness:** 3
**Presentation:** 3
**Contribution:** 3
**Rating:** 6
**Confidence:** 4

**Summary:**

Typically, training retrieval-augmented models with RL requires querying a retriever (or, say, a Web search API) for tens or hundreds of thousands of times. The authors argue that this can be prohibitive and that the lack of control over the retrieved document quality (i.e., the fact that even excellent search queries can fail to retrieve good documents) pose challenges in practice.

To tackle this, the authors introduce a method in which a small LLM can be trained (via behavior cloning / SFT) to imitate the behavior of the search engine API, at a low cost, and in a way that permits controlling the 'quality' of the retrieved documents (i.e., noisy vs. relevant). Using this search engine replacement at training time allows the authors to initiate the RL of the query generator with high-quality (fake) search results and to progressively degrades this quality. It also lowers the cost of calling Web search APIs.

The authors find that RL training with this module works well in practice, and even outperforms related work that trains directly using the search APIs.

**Strengths:**

1) The problem explored in this work is relatively novel. While this may not be the first time that someone trains a model to imitate a system or API, it is new to my knowledge in this domain (e.g., for RL towards RAG).

2) The ideas and presentation are simple and easy to follow. The results are compared against several recent related papers, and it is easy to appreciate from them that the core of this methods 'works' on the benchmarks selected.

**Weaknesses:**

Weakness 1:

The authors offer surprisingly little insight into why or how this method works at all. In the general case, it is simply not reasonable to expect that one can imitate an arbitrary search API with imitation learning. This is because there's, in principle, irreducible information in the search engine's knowledge of the world or certain domains that are simply not available in the model's pre-training data.

This argument does not imply that the method proposed in this paper will fail, because perhaps all that's needed for successful RL is that the questions in the corpus are answerable by the model or perhaps even just that the shape and format of the API responses are reasonable.

But we see from Table 3 that direct answer (or CoT) with the model itself has very low accuracy, i.e. the model seemingly has limited knowledge of these topics (at least when they're not decomposed). How do the authors explain this? What does the LLM-based search module actually output in practice; how correct or incorrect are its answers and how does that factor into the RL reward?

To me, this is a concerning weakness. As a reader, I lack a reason to accept how or why this method can work in the first place, which puts more weight behind Weakness 2 below.


Weakness 2:

The benchmarks used like NQ, TriviaQA, etc, while consistent with some recent papers, are all likely contaminated, old, and too easy. (There's no issue in including these tasks; what I take issue with is the absence of anything else.) I have to wonder whether the true explanation for weakness 1 is something along the lines of "the models know a whole lot about the topics in these datasets".

For example, the literature on HotPotQA from 2018-2021 would routinely achieve scores like 45-65%, even using much smaller and fundamentally weaker models, like BERT. Of course, much of this can be attributed to searching a Wikipedia dump versus searching the Web with a Web Search API, but the main point here is that with Weakness 1 in place, it's hard to understand how interesting a 34% on this benchmark is supposed to be. Aren't at least 30% of the questions in the benchmark answerable by the models (or even memorized by the models?) with some multi-hop decomposition?

Ultimately, I think Weaknesses 1 and 2 should be taken together holistically. I would be somewhat less concerned about the benchmarks if the _mechanism_ or even _patterns_ by which the method works were understood.

**Questions:**

See Weakness 1.

---

> ### Author Response · Authors · 2025-11-17
>
> We sincerely thank the reviewer for the careful reading and the insightful comments. We address the concerns point-by-point below.
>
> ### 1. **Why and how the proposed method works**
>
> We agree that understanding the mechanism is important. Our method works in this setting due to several properties of the simulation task:
>
> **(a) The search API we simulate is relatively simple.**
>  Unlike full-fledged web search results with long documents, heterogeneous formats, or latent ranking signals, we simulate only short snippets. This significantly reduces the complexity of the imitation task. As long as the LLM has the relevant knowledge and can mimic the response structure, it can generate search-like outputs that are close enough to those from the real API.
>
> **(b) Style/format imitation is handled by simulation tuning.**
>  We explicitly tune the small LLM to reproduce the “shape” of the search responses. This is a relatively easy behavior cloning problem, and we observe that even small models can learn the output format reliably.
>
> **(c) Knowledge limitations are mitigated through the design of the simulation prompt.**
>  We fully acknowledge that LLMs do *not* possess the same knowledge as a real search engine, and that the low direct-answer/CoT accuracy in Table 3 illustrates such limitations. To address this, our simulation LLM receives the question–answer pair as additional context, enabling it to generate snippets containing the *relevant information*, even when that information is not stored verbatim in its parameters. This greatly expands the effective knowledge boundary of the simulated search engine.
>
> Overall, while we fully agree that perfectly imitating an arbitrary real-world search API is infeasible, our problem setting is intentionally scoped such that the simulation is practical and useful for RL. We view this work as an initial step in this direction, and expect that continued advances in LLM capabilities will further improve the fidelity and generality of such simulated search environments.
>
> ### 2. **Benchmark selection and contamination concerns**
>
> We agree that benchmarks such as NQ and TriviaQA are relatively old and may contain contamination—a limitation shared by many prior works in this area. Importantly, our goal in this paper is to evaluate the *relative* effectiveness of our method under strictly controlled and identical experimental conditions.
>
> As shown in Table 3, when keeping all factors fixed (datasets, search engine, model size, and evaluation pipeline), our approach consistently outperforms strong baselines, including direct answer, CoT, and standard RAG/search-agent systems that rely on real search APIs. These controlled comparisons allow us to fairly evaluate the effectiveness of our method even in the presence of potential dataset contamination.
>
> Once again, we thank the reviewer for the thoughtful analysis. We hope these clarifications address the key concerns and help make the mechanisms and contributions of our work clearer.

---

### Note · Program_Chairs · 2026-01-17
**Submission Desk Rejected by Program Chairs**

The following references in this submission do not refer to real documents and/or have major errors in bibliographic information:

 Varun Kumar, Le Hou, Yi Guo, Rui Shao, Peter Griffin Jin, Wei Weng, Yujia Xie, and Tie-Yan Liu. Self-correcting language models with reinforcement learning. arXiv preprint arXiv:2409.06543, 2024.
Ritesh Kumar and et al. Research: Autonomous retrieval decision-making in llms using reinforcement learning. In $I C L R, 2025$.